# The Effect of HbA_1c_ Variability as a Risk Measure for Microangiopathy in Type 1 Diabetes Mellitus

**DOI:** 10.3390/diagnostics11071151

**Published:** 2021-06-24

**Authors:** Pedro Romero-Aroca, Raul Navarro-Gil, Albert Feliu, Aida Valls, Antonio Moreno, Marc Baget-Bernaldiz

**Affiliations:** 1Ophthalmology Service, University Hospital Sant Joan, Institut de Investigacio Sanitaria Pere Virgili (IISPV), Universitat Rovira & Virgili, 43204 Reus, Spain; raul_navarro_gil@hotmail.com (R.N.-G.); mbaget@gmail.com (M.B.-B.); 2Pediatric Service, University Hospital Sant Joan, Institut de Investigacio Sanitaria Pere Virgili (IISPV), Universitat Rovira & Virgili, 43204 Reus, Spain; albert.feliu.rovira@gmail.com; 3Departament d’Enginyeria Informàtica I Matemàtiques, Escola Tècnica Superior d’Enginyeria, Universitat Rovira & Virgili, ITAKA-Intelligent Technologies for Advanced Knowledge Acquisition, 43204 Tarragona, Spain; aida.valls@urv.cat (A.V.); antonio.moreno@urv.cat (A.M.)

**Keywords:** HbA_1c_ variability, coefficient of variation of HbA_1c_, diabetic retinopathy, severity of diabetic retinopathy

## Abstract

Background: To measure the relationship between variability in HbA_1c_ and microalbuminuria (MA) and diabetic retinopathy (DR) in the long term. Methods: A prospective case-series study, was conducted on 366 Type 1 Diabetes Mellitus patients with normoalbuminuria and without diabetic retinopathy at inclusion. The cohort was followed for a period of 12 years. The Cox survival analysis was used for the multivariate statistical study. The effect of variability in microangiopathy (retinopathy and nephropathy) was evaluated by calculating the standard deviation of HbA_1c_ (SD-HbA_1c_), the coefficient of variation of HbA_1c_ (CV-HbA_1c_), average real variability (ARV-HbA_1c_) and variability irrespective of the mean (VIM-HbA_1c_) adjusted for the other known variables. Results: A total of 106 patients developed diabetic retinopathy (29%) and 73 microalbuminuria (19.9%). Overt diabetic nephropathy, by our definition, affected only five patients (1.36%). Statistical results show that the current age, mean HbA_1c_, SD-HbA_1c_ and ARV-HbA_1c_ are significant in the development of diabetic retinopathy. Microalbuminuria was significant for current age, mean HbA_1c_, CV-HbA_1c_ and ARV-HbA_1c_. Conclusions: By measuring the variability in HbA_1c_, we can use SD-HbA_1c_ and ARV-HbA_1c_ as possible targets for judging which patients are at risk of developing DR and MA, and CV-HbA_1c_ as the target for severe DR.

## 1. Introduction

It is estimated that 415 million people worldwide were living with some form of diabetes in 2015 [1] and that number has been predicted to rise to around 640 million by 2040 [2]. It has become a chronic disease with several complications. Diabetes Mellitus (DM) is classified as Type 1 Diabetes (T1DM), Type 2 Diabetes (T2DM), gestational diabetes (GDM), monogenic diabetes (MODY) and secondary diabetes [3]. There is a current trend towards children developing T1DM and more than half a million children are estimated to be currently living with the disease.

Diabetes Mellitus (DM) is one of the main causes of morbidity and mortality in the developed world. Microangiopathy affects the retinal vessels and leads to diabetic retinopathy (DR), which is a major case of visual loss worldwide [4]. Furthermore, the effect on kidneys leads to overt nephropathy (ON). Currently, we know that an early form of kidney damage in DM1 patients is microalbuminuria (MA), which is an early-stage diabetic nephropathy.

It is well established that chronic hyperglycaemia is one of the main risk factors for microangiopathy and can be assessed by determining HbA_1c_ levels [5].

However, in clinical practice, we observe that patients with good HbA_1c_ levels can still develop DR or nephropathy. Some authors pointed to the likelihood of another factor, independent from the updated mean HbA_1c_, that might contribute to the risk of developing complications, suggested as glycaemic variability (GV) [6]. Initial studies attempted to evaluate the intra-day and inter-day changes in glycaemia, but with no success [7,8,9]. Since then, studies of long-term changes in glycaemia, measured by changes in HbA_1c_, have been successful [10,11,12,13,14,15,16]. Long term changes in glycaemia can currently be calculated by measuring the mean and standard deviation of HbA_1c_ (SD-HbA_1c_), the coefficient of variation of HbA_1c_ (CV-HbA_1c_), the average real variability (ARV-HbA_1c_) and the variability independent of the mean (VIM-HbA_1c_) [17,18].

The aim of present study was to measure the relationship between diabetic retinopathy and diabetic nephropathy development according to variability in HbA_1c_, measured by the following parameters: mean, SD-HbA_1c_, CV-HbA_1c_, AVR-HbA_1c_ and VIM-HbA_1c_.

## 2. Subjects

### 2.1. Setting

The reference population in our area is 247,174. The total number of DM patients registered with our Health Care Area (University Hospital Saint Joan, Tarragona, Spain) is 17,792 (7.1%). Our DR screening programme has been ongoing since 2007, when we offered a retinography annually to our T1DM patients. The screening programme is described more completely elsewhere [19,20].

### 2.2. Design

We carried out a prospective, case series study of 366 T1DM patients who were not part of the screening programme on 1 January 2007 and who did not initially have DR or MA.

Inclusion criteria: T1DM patients had to have a minimum of eight prior HbA_1c_ measures.

Exclusion criteria: patients with T2DM, GDM or other specific types of diabetes due to other causes, e.g., monogenic diabetes syndromes (such as neonatal diabetes and MODY), diseases of the exocrine pancreas (such as cystic fibrosis and pancreatitis), and drug—or chemical—induced diabetes (such as with glucocorticoid use in the treatment of HIV/AIDS, or after organ transplantation).

## 3. Material and Methods

A total of 366 patients with T1DM, as diagnosed by endocrinologists, were followed between 1 January 2007 and 31 December 2020. The retinographies were taken under mydriasis according to the Joslin Vision Network with 3 fields of 45ª (macula-focused; nasal; temporal superior) using a Topcon NW400 retinal camera, [21]. All patients were submitted to one retinography per year.

The DR was diagnosed by reading the retinographies by a retinal expert ophthalmologist, and diagnosis was determined when microaneurysms were present in the fundus retinography. Classification was conducted according to the International Council of Ophthalmology, ICO [22] as (i) mild DR with only microaneurysms, (ii) moderate DR (microaneurysms, hard exudates, haemorrhages and venous abnormalities), (iii) severe DR (the above together with one of the following: >20 haemorrhages in each quadrant, venous anomalies defined in 2 quadrants, intra-retinal microvascular abnormalities in 1 quadrant, no signs of proliferation, and (iv) proliferative DR, defined by a presence of neovascularization.

In patients diagnosed with diabetic retinopathy, a complete ophthalmological examination was performed that included visual acuity, anterior segment biomicroscopy and optical coherence tomography (OCT).

General Practitioners and endocrinologists provided information on the duration of DM, arterial hypertension and body mass index (BMI).

### 3.1. Laboratory Analysis

A venous blood sample was obtained after fasting and serum and EDTA plasma were stored at −80 °C until measurements were taken.

Levels of glycosylated haemoglobin (HbA_1c_) [19] were measured at least twice a year, as recommended by the American Diabetes Association, during the 12-year follow-up and a mean of all values was applied to the study. Glycaemia was controlled according to the European Diabetes Policy Group, and the standards of medical care in diabetes of the American Association of Diabetes [23,24]. The HbA_1c_ values were obtained after blood extraction and were standardized according to the DCCT reference range (20.7–42.6 mmol/mol) [25]. The mean HbA_1c_ (Mean) values were calculated after a minimum of 8 HbA_1c_ determinations per patient in the four years prior to DR diagnosis or the last visit.

The urine analysis was performed at least once per year and the presence of MA, defined as an increased albumin excretion of 30–300 mg/g (30–300 mg of albumin/ 24 h or 20–200 μg/min of albumin) in two out of three tests repeated at intervals of 3–6 months, as well as exclusion conditions that invalidate the test. Following a diagnosis of MA, there was repeat testing over a period of 3–4 months. Presence of overt nephropathy, defined as both clinical albuminuria or overt nephropathy by the American Diabetes Association, corresponding to protein excretion >300 mg/24 h. Glomerular filtration rate (eGFR), as measured by the chronic kidney disease epidemiology collaboration equation CKD-EPI, was estimated on the same urine collection. The microalbuminuria development was defined as MA onset during the studied period.

Finally, we determined the serum levels of the following: HDL cholesterol, LDL cholesterol and triglycerides. In the statistical analysis, we classified patients into normal or the following risk levels: HDL cholesterol normal value ≥1.10 mg/dL, LDL cholesterol normal value ≤2 mg/dL. Triglycerides normal value ≤1.70 mg/dL, at least one year determination of lipid profile was performed.

### 3.2. Variability of HbA_1c_ Calculation

Variability of HbA_1c_ was measured by four different values:The standard deviation of the mean HbA_1c_ (SD-HbA_1c_).The average real variability (ARV-HbA_1c_) is the average of the absolute differences between consecutive HbA_1c_ measurements.The coefficient of variation of HbA_1c_, (CV-HbA_1c_) applying the following formula, [12] CV-HbA_1c_ = SD-HbA_1c_/Mean HbA_1c_The variability independent of the mean (VIM-HbA_1c_) is a transformation of the standard deviation, which is not correlated with mean HbA_1c_ and is calculated as follows [26]:
VIM-HBA_1c_ = k x tandard deviation of HbA_1c_ (SD-HbA_1c_)/Mean (HbA_1c_) ^x^
where x is calculated from fitting a power model: SD-HbA_1c_ = constant x Mean HbA_1c_ ^x^ and k = Mean (Mean HbA_1c_) ^x^.

### 3.3. Statistical Methods

Dependent variables were DR and MA, and the independent variables were current age, gender, duration of DM, arterial hypertension, body mass index (BMI), the mean-HbA_1c_, lipid profile (determining LDL cholesterol, HDL cholesterol and triglycerides), renal status (estimated glomerular filtration rate (eGFR) as measured by the chronic kidney disease epidemiology collaboration equation CKD-EPI).

Variability of HbA_1c_ was measured by the following parameters: SD-HbA_1c_, CV-HbA_1c_, ARV-HbA_1c_ and VIM-HbA_1c_.

Data were evaluated and analysed using the SPSS 22.2 statistical software package and statistical significance was set at *p* < 0.05. The specific statistical study carried out and the specific type of tests applied depended on the data obtained and their distribution. Descriptive statistical data determined the mean, standard deviation, minimum and maximum values Student’s t-test was applied for independent samples. The inferential analysis was carried out through the creation of contingency tables, and the chi-squared test for qualitative variables. In cases where the reliability of this test was not guaranteed, we used Fisher’s exact test. The two proportions in paired samples were compared using the McNemar test. The different time-dependent variables were compared to know their influence on the development of DR and MA by applying survival analysis using the Cox Proportional Hazards regression model.

## 4. Results

### 4.1. Demographic Variables of Sample Size

From 2007 to 2019, 366 T1DM patients were studied. Sample characteristics at the end of the study were as follows: current age = 35.20 ± 10.03 years, 193 = males (52.7%) and 173 = females (47.3%).

A total of 106 patients developed DR (29%) and 73 MA (19.9%). Overt diabetic nephropathy, by our definition, affected only five patients (1.36%), and with such a small number of patients we did not carry out any statistical analysis.

By DR classification at the end of the study, we observed 70 patients (19.1%) with mild DR, 20 patients (5.5%) with moderate DR, eight patients (2.2%) with severe DR and five patients (1.4%) with proliferative DR. Table 1 shows the differences in the other parameters between groups and their significant values.

### 4.2. Univariate Analysis of Diabetic Retinopathy

Table 1 shows the univariate analysis, in which current age, arterial hypertension, DM duration, eGFR and mean-HbA_1c_ were significant risk factors. For variability, all studied parameters (SD-HbA_1c_, CV-HbA_1c_, ARV-HbA_1c_ and VIM-HbA_1c_) were significant, despite VIM-HbA_1c_ being the only one with significance above 0.001 and with a value of *p =* 0.037.

### 4.3. Univariate Study of the Severity of Diabetic Retinopathy

The univariate analysis shows significant differences in DR for current age (*p* < 0.001), DM duration (*p* < 0.001), arterial hypertension (*p* < 0.001), and mean HbA_1c_ (*p* < 0.001). For HbA_1c_ variability, all four studied parameters were significant: SD-HbA_1c_ (*p* < 0.001), VC-HbA_1c_ (*p* < 0.001), VIM-HbA_1c_ (*p* < 0.001), and ARV-HbA_1c_ (*p* < 0.001) (Table 2).

### 4.4. Microalbuminuria Univariate Analysis

Table 3 shows the univaritate analysis, in which current age, arterial hypertension, DM duration, meanHbA_1c_ and eGFR were all significant risk factors. Variability in SD-HbA_1c_, CV-HbA_1c_ and ARV-HbA_1c_ was significant, but VIM-HbA_1c_ was not significant at *p =* 0.750. The relationship between MA and DR in this study are significant at a *p =* 0.003, despite the previous analysis (Table 3) for the presence of DR of microalbuminuria not being significant. At this point, we should explain that the statistical study was different in that we used chi-squared (a qualitative test) for MA and we used Student’s t-test (a quantitative test) for DR.

### 4.5. Multivariate Study of Diabetic Retinopathy

For the survival study of DR, we used the Cox Proportional Hazards model that determines which variables are significant in the development of DR with the duration of DM as a time variable. Our results determined that age at the end of the study and the metabolic control of DM measured by mean-HbA_1c_ values were significant for DR, and the variability parameters of SD-HbA_1c_ A1c and ARV-HbA_1c_ were significant (Table 4).

### 4.6. Multivariate Study of Severity of Diabetic Retinopathy

Applying the Cox survival analysis for the severity of DR, only current age (HR = 1.159, *p =* 0.003) and mean-HbA_1c_ (HR = 1.321, *p =* 0.002) were significant and for the HbA_1c_ variability study, the SD-HbA_1c_ with HR = 1.514 *p* < 0.001 and CV-HbA_1c_ with HR = 1.290, *p =* 0.012 were significant.

The MA study does not show any link with severity of DR in the Student’s analysis (*p* = 0.739) (Table 4).

### 4.7. Survival Analysis of Microalbuminuria

For the survival study of MA using the Cox Proportional Hazards model, our results were: current age (HR = 1.957, *p* = 0.008) and mean-HbA_1c_ (HR = 1.472, *p* = 0.026), significant for MA.

In the HbA1c variability study, the SD-HbA_1c_ (HR 1.377, *p* = 0.028) and ARV-HbA_1c_ (HR 1.179, *p* = 0.036) were significant (Table 4).

## 5. Discussion

The results of the present study show that current age is a risk factor in the development of DR and MA, which might be due to the duration of DM, a well-known risk factor in the development of microangiopathy. In addition, other parameters such as arterial hypertension, eGFR measured through CKD-EPI and the mean of HbA1c levels were all significant for both forms of microangiopathy. All these parameters are well-known risk factors for DR and developing of MA, as the DCCT and the EDIC also reported following their extensive studies [27]. Among the different parameters that measure the importance of variability in HbA_1c_ in microangiopathy, the univariate statistical analysis determined that SD-HbA_1c_, CV-HbA_1c_, VIM-HbA_1c_ and ARV-HbA_1c_ were all significant for DR and MA. However, the results changed following adjustments for other significant variables, such as current age, arterial hypertension, eGFR and mean-HbA_1c_. For developing DR, only SD-HbA_1c_ (HR 1.966, *p =* 0.018) and ARV-HbA_1c_ (2.171, *p =* 0.002) were significant. Moreover, for microalbuminuria, only SD-HbA_1c_ (HR 1.377, *p =* 0.0128) and ARV-HbA_1c_ (1.179, *p =* 0.036) were significant. Only for DR severity did the CV-HbA_1c_ become significant (HR =1.290, *p =* 0.012). VIM-HbA_1c_ was not significant for DR, MA or DR severity.

If we compare our results to other studies on HbA_1c_ variability, there have been three that have studied only DR development [13,15,16], three that have studied only nephropathy [11,12,14], and one that has studied both forms of microangiopathy [10]. The most important study was by Herman et al. [15], a retrospective study focusing on the CV-HbA_1c_ of over 35,891 T1DM patients. Variability in HbA_1c_ was reported as a risk factor for DR, independent of metabolic control, with HR 1.11 at ten years of DM duration.

The other two studies on HbA_1c_ variability and DR both focused on severe forms of DR. Hietala et al. [13] was a retrospective study over a 5.5-year follow-up with a sample size of 2029 T1DM patients, which reported a relationship between CV-HbA_1c_ and laser treatments of patients. The second study, by Schreuer et al. [16], was a cross-sectional study on 415 T1DM patients, which also demonstrated a relationship between CV-HbA_1c_ and patients who develop sight-threatening DR (HR 1.054). The present study did not demonstrate any relationship between CV-HbA_1c_ and DR development, but a positive relationship with severe DR, similar to the Hietala and Schreuer studies. Herman adjusted the results for age and gender, and we adjusted our results for DM duration, renal status (using eGFR and MA) and other parameters, such as arterial hypertension and lipid profile, differences which might explain our results. Another group of studies that we compare are those on MA and diabetic nephropathy. The first study is by Marcoveccchio et al. [12], who found that 438 patients from a sample of 1232 patients with DM1 had a positive relationship between SD-HbA_1c_ and MA with HR 1.04, and also a positive relationship between SD-HbA_1c_ and MA with HR 1.31. The second study was by Nazim et al. [14], a cross-sectional study that also found a positive relationship between MA and SD-HbA_1c_ (HR 1.04), and a third study by Waden et al. [11], a prospective 5.7-year follow-up of 2107 T1DM patients again found a positive relationship between SD-HbA_1c_ and a progression in renal status (HR 1.92).

The last study to discuss is by Kilpatrick et al. [10], who applied data from the DCCT to 1441 T1DM patients, checking whether long-term variability had any effect on the development of DR. Their results showed that long-term variability, measured by the SD-HbA_1c_, increased the risk of developing both DR (HR 2.26), and nephropathy (SD-HbA_1c_ HR 1.86).

Our results are similar to those published in the literature, with some variations. We found a positive relationship between SD-HbA_1c_ and DR development and microalbuminuria. The CV-HbA_1c_ parameter is only significant for severe DR, in agreement with Hietala and Screuer but contradicting Herman. Regarding the other studied parameters, VIM-HbA_1c_ and ARV-HbA_1c_, there have been no published results for T1DM patients, but in T2DM, reported by Takao et al. [28], independent of the mean (VIM) of HbA_1c_ and of systolic blood pressure the variation can predict the appearance of DR and MA. In the present study this parameter was not significant for DR or MA. The VIM is used in the control of arterial hypertension proving its effectiveness for SBP variability [17], but in current study we studied the association of arterial hypertension and not SBP values, this may be the cause of the lack of statistical significance in our study.

We can conclude that VIM was not important for detecting HbA_1c_ variability in T1DM patients or elucidating its relationship with DR development.

Finally, ARV-HbA_1c_ is a parameter that has not been studied in T1DM patients. It is closely related to mean-HbA_1c_, which is a well-known risk factor for DR and MA; therefore, we can conclude that AR-HbA_1c_ might help us to determine DR development. Regarding the relationship between both microangiopathies, MA and DR, our study group has published some articles previously [29,30] with similar results to the present study. MA is not a risk factor for DR development but the presence of DR can be a marker of the presence of MA in T1DM patients.

This study has some limitations. It is retrospective and the sample of T1DM patients is small, with only 366 patients who met the inclusion criteria; therefore, an extrapolation of our results to other populations still needs to be demonstrated. Another limitation is that it only takes account of HbA_1c_-variability patients with a minimum of eight previous HbA_1c_ values. Therefore, we need more studies with a longer follow-up period in order to obtain more useful data for clinical practice.

The strengths of our study are, firstly, that the sample size is in fact highly representative of our population as a whole because patients were recruited from our own T1DM screening programme, and secondly, we have included all the risk variables that might influence DR or MA.

## 6. Conclusions

In conclusion, long-term glycaemic variability emerges as a target that needs to be corrected in order to avoid complications in Diabetes Mellitus, such as diabetic retinopathy. The standard deviation of HbA_1c_ and average real variability of HbA_1c_ are better related to diabetic retinopathy and microalbuminuria and it will be a possible variable for detecting patients at risk of developing microangiopathy. The coefficient of variation of HbA_1c_ was related to sever DR and we have not demonstrated any significance of the variation independent of the mean (VIM-HbA_1c_) with DR nor MA development. More studies with stronger evidence and a longer follow-up period are essential if we are to obtain better data for clinical practice.

## Figures and Tables

**Table 1 diagnostics-11-01151-t001:** Univariate study of diabetic retinopathy.

Variable	Without Diabetic Retinopathy	With Diabetic Retinopathy	Significance
Current age (years)	35.87 ± 10.22	42.47 ± 8.76	*p =* 0.026
Male (%)	134 (51.53)	59 (55.66)	*p =* 0.181
Arterial hypertension (%)	13 (3.55)	27 (21.58)	*p* < 0.001
DM duration (years)	15.17 ± 8.3	20.92 ± 9.51	*p =* 0.034
LDL cholesterol (mg/dL)	101.33 ± 27.71	103.83 ± 25.48	*p =* 0.674
HDL cholesterol (mg/dL)	75.27 ± 18.04	60.9 ± 18.93	*p =* 0.386
Triglycerides (mg/dL)	108.73 ± 14.25	104.02 ± 15.35	*p =* 0.213
Microalbuminuria (mg/g)	17.49 ± 11.26	31.15 ± 14.27	*p =* 0.151
eGFR (mL/min/1.73 m^2^)	106.75 ± 15.77	96.11 ± 18.67	*p =* 0.003
Mean-HbA_1c_ (%) mmol/mol	7.56 ± 0.88 59.12 ± 13.87	8.86 ± 1.44 73.33 ± 7.75	*p* < 0.001
Variability HbA1c data			
SD-HbA_1c_	0.45 ± 0.36	1.18 ± 0.67	*p* < 0.001
CV-HbA_1c_	0.058 ± 0.047	0.112 ± 0.079	*p* < 0.001
ARV-HbA_1c_	0.78 ± 0.59	2.09 ± 0.98	*p* < 0.001
VIM-HbA_1c_	0.38 ± 0.07	0.41 ± 0.06	*p =* 0.037

**Table 2 diagnostics-11-01151-t002:** Differences in severity of diabetic retinopathy with significant variables and microalbuminuria.

	Mild DR	Moderate DR	Severe DR	Proliferative DR	Significance
Current age (years)	37.86 ± 10.19	42.55 ± 9.32	42.56 ± 8.92	46.01 ± 7.91	*p* < 0.001
Arterial hypertension (%)	17 (24.28)	6 (30)	4 (50)	4 (80)	*p* < 0.001
Diabetes duration (years)	17.24 ± 8.26	18.24 ± 8.75	20.62 ± 9.6	27.8 ± 8.37	*p* < 0.001
Mean-HbA_1c_ (%) (mmol/mol)	8.68 ± 1.43 71.36 ± 7.86	9.16 ± 1.22 76.61 ± 10.16	9.86 ± 1.25 84.26 ± 9.83	10.36 ± 1.45 89.72 ± 7.65	*p* < 0.001
Study of variability					
SD-HbA_1c_	1.05 ± 0.5	1.37 ± 0.79	1.82 ± 0.85	1.91 ± 1.18	*p* < 0.001
CV-HbA_1c_	0.101 ± 0.069	0.126 ± 0.087	0.175 ± 0.107	0.186 ± 0.117	*p* < 0.001
VIM-HbA_1c_	0.40 ± 0.06	0.41 ± 0.04	0.47 ± 0.05	0.51 ± 0.04	*p* < 0.001
ARV-HbA_1c_	1.86 ± 0.92	2.51 ± 0.92	2.75 ± 1.12	2.66 ± 1.04	*p* < 0.001
Study of microalbuminuria					
Microalbuminuria (mg/g)	37.2 ± 19.9	16.8 ± 18.11	18.23 ± 17.62	37.92 ± 17.11	*p =* 0.739

**Table 3 diagnostics-11-01151-t003:** Univariate study of microalbuminuria.

Variable	Without Microalbuminuria	With Microalbuminuria	Significance
Current age (years)	36.6 ± 10.25	41.58 ± 8.76	*p =* 0.034
Male (%)	152 (51.87)	41 (56.16)	*p =* 0.102
Arterial hypertension (%)	25 (8.53)	15 (20.54)	*p* < 0.001
DM duration (years)	15.96 ± 8.76	20.41 ± 9.18	*p =* 0.029
LDL cholesterol (mg/dL)	102.23 ± 26.55	101.32 ± 29.26	*p =* 0.552
HDL cholesterol (mg/dL)	58.69 ± 17.04	61.36 ± 19.13	*p =* 0.255
Triglycerides (mg/dL)	106.53 ± 13.68	110.73 ± 16.27	*p =* 0.509
eGFR (mL/min/1.73 m^2^)	105.33 ± 16.16	97.16 ± 20.16	*p =* 0.001
Diabetic retinopathy (%)	50 (17.1)	56 (76.7)	*p =* 0.003
Mean-HbA_1c_ (%) mmol/mol	7.76 ± 1.13 61.85 ± 10.92	8.79 ± 1.36 66.01 ± 8.63	*p =* 0.003
Variability HbA1c data			
SD-HbA_1c_	0.54 ± 0.43	1.15 ± 0.8	*p* < 0.001
CV-HbA_1c_	0.062 ± 0.048	0.117 ± 0.091	*p* < 0.001
ARV-HbA_1c_	0.95 ± 0.74	2.01 ± 1.13	*p* < 0.001
VIM-HbA_1c_	0.38 ± 0.07	0.41 ± 0.07	*p =* 0.750

**Table 4 diagnostics-11-01151-t004:** Survival study of diabetic retinopathy.

**Diabetic Retinopathy**		
Variable	Hazard Ratio (95% CI)	Significance
Current age (years)	1.955 (1.57–2.528)	*p* < 0.001
Arterial hypertension	1.149 (0.646–2.044)	*p =* 0.635
eGFR (mL/min/1.73 m^2^)	0.999 (0.988–1.011)	*p =* 0.913
Mean-HbA_1c_	3.502 (1.081–11.349)	*p =* 0.037
SD-HbA_1c_	1.966 (1.125–3.434)	*p =* 0.018
CV-HbA_1c_	1.448 (0.897–2.456)	*p =* 0.169
ARV HbA_1c_	2.171 (1.326–3.555)	*p =* 0.002
VIM-HbA_1c_	0.672 (0.397–1.130)	*p =* 0.134
**Diabetic retinopathy severity ***		
Variable	Hazard ratio (95% CI)	Significance
Current age (years)	1.159 (1.057–2.991)	*p =* 0.003
Arterial hypertension	1.479 (0.842–2.597)	*p =* 0.173
eGFR (mL/min/1.73 m^2^)	1.001 (0.989–1.012)	*p =* 0.928
Mean-HbA_1c_	1.321 (1.108–1.575)	*p =* 0.002
SD-HbA_1c_	1.744 (1.089–3.385)	*p* < 0.001
CV-HbA_1c_	1.390 (1.076–1.796)	*p =* 0.012
ARV- HbA_1c_	0.514 (0.002–1.893)	*p =* 0.809
VIM-HbA_1c_	0.100 (0.005–1.912)	*p =* 0.126
**Microalbuminuria**		
Variable	Hazard ratio (95% CI)	Significance
Current age (years)	1.957 (1.357–2.787)	*p =* 0.008
Arterial hypertension	1.049 (0.412–1.735)	*p =* 0.892
eGFR (mL/min/1.73 m^2^)	1.002 (0.760–1.014)	*p =* 0.734
Mean-HbA_1c_	1.472 (1.029–1.572)	*p =* 0.026
SD-HbA_1c_	1.377 (1.006–3.554)	*p =* 0.028
CV-HbA_1c_	1.025 (0.461–2.278)	*p =* 0.952
ARV HbA_1c_	1.179 (1.020–1.864)	*p =* 0.036
VIM-HbA_1c_	0.264 (0.208–4.252)	*p =* 0.449

Diabetic retinopathy severity * = moderate DR + severe DR + proliferative DR.

## Data Availability

The database used and analysed is available from the corresponding author on research request.

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
