# Peer review of "The Effect of HbA1c Variability as a Risk Measure for Microangiopathy in Type 1 Diabetes Mellitus"

_diagnostics, 2021, doi:10.3390/diagnostics11071151_

Round 1

Reviewer 1 Report

In this interesting paper, the authors investigated the associations between HbA1c variability and the development of diabetic retinopathy (DR) and microalbuminuria (MA) in a relatively large cohort of 366 type 1 diabetes mellitus patients, who did not initially have DR and MA. 

In the present study, SD-HbA1c was associated with DR and microalbuminuria on multivariate analysis. ARV-HbA1c was similarly associated with both, but CV-HbA1c and VIM-HbA1c were neither associated with DR nor with microalbuminuria. HbA1c mean values had a stronger association with DR and  microalbuminuria than HbA1c variability parameters in the multivariate models. I do feel that this manuscript adds to the existing knowledge about the relationship between ocular and systemic markers in diabetic eye disease. 

However, in my opinion the manuscript can be improved further with minor edits, please find below some comments and suggestions that the authors may wish to address to improve even more the quality of the manuscript.

1 - Abstract and Methods
Authors said that "A prospective population based study on type 1 diabetes"... Studies on type 1 diabetes are essentially not population based, therefore I would suggest to rephrase this sentence of the abstract. 

2 - Methods

- Was progression of DR also considered an outcome besides DR onset?  

- Was the progression from microalbuminuria to clinical nephropathy also considered an outcome?

- Regarding microalbuminuria development/progression, what was the criteria employed to define Hba1c variability as a predictor of its occurrence or progression? I think it was the same criterion used for DR outcome (that is, having 8 HbA1c measurements and no microalbuminuria onset during this period). The authors should clarify this aspect in the manuscript text.  

- Lipid parameters were obtained for this analysis only once, at which moment were they obtained? That is not clear in the text.

3.- Results

- I think that somehow the data presented in table 1 replicates the information given in the text and table 2, I would suggest to the authors to reconsider whether this table is necessary. 

- Table 2: I do feel that the text presenting results of table 2 should be rewritten, as it is important to comment that VIM-HbA1c was the least significant parameter in the intergroup comparisons.

- Table 3: I think it is more meaningful to divide data only in two columns for the intergroup comparison (mild DR vs  moderate or higher DR stage) because of the small number of cases with advanced levels of DR (moderate, severe and proliferative diabetic retinopathy).

- Table 5 results: How were the individuals grouped according to DR severity?

- Reference values for lipids are expressed in mmol/L in the methods section   but detailed as mg/dL in the results section. Please edit this and use only one  method for consistency.  

- Albumin/creatinine ratio should be expressed as g/mol.

Author Response

Responses to reviewer 1

I appreciate the reviewer comments, and the time s/he has devoted to this study.

1 - Authors said that "A prospective population-based study on type 1 diabetes " Studies on type 1 diabetes are not population based.

Certainly, this is not a population study, we changed the paragraph in abstract and methods by: “We carried out a prospective, case-series study.”

  1. Was progression of DR also considered an outcome? Was the progression from microalbuminuria to clinical nephropathy also considered an outcome?

Yes, both progression of diabetic retinopathy and diabetic nephropathy were considered outcomes.

  1. Regarding microalbuminuria development/progression what was the criterion employed to the individuals be included in the evaluation of Hba1c variability as a predictor of its occurrence or progression? I think it was the same criterion used for DR outcome (that is, having 8 Hba1c measurements and no microalbuminuria onset during this period). The authors should clarify this aspect in the text.

We clarified this point in methods with following sentence: “The microalbuminuria development was defined as MA onset during the studied period.”

  1. Lipid parameters were obtained for this analysis only once, at which moment where they obtained? That is not clear in the text.

We included in methods the sentence: “at least one year determination of lipid profile was performed.” That we think clarify this point.

  1. I think that data in table 1 are not necessary it is well explained in text and table 2

We eliminated table 1 and only table 2 was included in new version renamed as table 1.

  1. Table 2 the text presenting results of table 2 should be rewritten, it is important to comment that VIM-Hb1c was the one with a less significant difference in intergroup comparison.

We included a sentence in results text about this point. “VIM-HbA1c being the only one with significance above 0.001 and with a value of p= 0.037.”

  1. Table 3: I think is better for group comparisons include only two columns mild Dr and moderate or higher DR because of the small number of moderate, severe and proliferative diabetic retinopathy.

Certainly, reviewer comments are correct, but we think that it is important for readers explain differences between different diabetic retinopathy severity groups.

  1. Table 5 results. How were the individuals grouped according to DR severity?

In table we have included the following sentence: Diabetic retinopathy severity* = moderate DR + severe DR + proliferative DR

  1. Reference values for lipids are expressed in mmol/L in methods but as mg/dL in results.

We changed reference values in methods to mg/dL as in results.

Reviewer 2 Report

The aim of the study is expressed in te abstract as follows: To measure the relationship between variability in HbA1c and microalbumi-15nuria (MA) and diabetic retinopathy (DR) in the long-term.

However, it is not mentioned how do you measure diabetic retinopathy or its development? In my opinion, the aim of this study was to measure the variability of HbA1c in patients with diabetes. Additionally, you checked the prevalence of diabetic retinoipathy in these patients. The aim of the study  should be modified. The conclusion should address the purpose of the study.

How often the photos of the funduse were taken during 13 years of the followup (time intervals) ?  Was the pupil dilated? Was visual acuity and OCT examinations performed?  Who assessed fundus pictures? Was only type I or type II diabetes included or both? The mean ag eis only 35 years, thus you studied young population of patients.

Author Response

Responses to reviewer 2

I appreciate the reviewer comments, and the time s/he has devoted to this study.

  1. We changed the aim of the study and have included the following sentence

The aim of present study was to measure the relationship between diabetic retinopathy and diabetic nephropathy development according to variability in HbA1c, measured by the fol-lowing parameters: Mean, SD-HbA1c, CV-HbA1c, AVR-HbA1c and VIM-HbA1c.”

  1. However, it is not mentioned how do you measure diabetic retinopathy or its development?

In new version we included the paragraph:

The DR was diagnosed reading the retinographies by a retinal expert ophthalmologist, and diagnosis was made when microaneurysms were present in the fundus retinography. Classification was made according to the International Council of Ophthalmology, ICO

  1. In response to reviewer doubt about the aim of the study We made changes in the text. Also in methods an expressed that the aim is the relationship between the diabetic retinopathy development and diabetic nephropathy development and the variability of HbA1c along study period.
  2. How often the photos of the fundus were taken during 13 years of the follow up?

The retinographies were performed at least one per year, we included in the text a sentence to explain this point.

  1. Was the pupil dilated?

Yes, the pupil was dilated in all patients. We included a sentence in methods about this point.

  1. Was visual acuity and OCT examinations performed?

Yes, the visual acuity was performed in all patients in each visit, and OCT only in patients with diagnosis of diabetic retinopathy.

  1. Who assessed fundus pictures?

Retinal specialist read all retinographies and performed Dr diagnosis. We included in methods a paragraph for this point:

The DR was diagnosed by reading the retinographies by a retinal expert ophthalmologist.

  1. Was only type 1 or type 2 diabetes included or both?

Only type 1 diabetes mellitus were included in the study in methods we specify this point:

Exclusion criteria: patients with T2DM,..”

  1. The mean age is 35 years, thus you studied young population of patients.

Certainly, the sample size is made up of patients with a young age, since we need patients without diabetic retinopathy and without diabetic nephropathy at the beginning of the study, which leads to the population having a mean age of young patients

Round 2

Reviewer 2 Report

All the comments have been adressed.